# Spontaneous Contralateral Adrenal Hemorrhage during Periduodenal Abscess Drain Placement

**DOI:** 10.3390/diagnostics14030334

**Published:** 2024-02-04

**Authors:** Eusha Hasan, Ashley Lamba, Abheek Ghosh, Hakob Kocharyan, Mustafa Al-Roubaie, Christopher Yeisley

**Affiliations:** 1Department of Vascular and Interventional Radiology, Donald and Barbara Zucker School of Medicine at Hofstra/Northwell, Hempstead, NY 11549, USA; ehasan1@pride.hofstra.edu; 2Department of Diagnostic Radiology and Nuclear Medicine, University of Maryland Medical Center, Baltimore, MD 21201, USA; abheek.ghosh@som.umaryland.edu; 3Department of Vascular and Interventional Radiology, Moffitt Cancer Center, Tampa, FL 33612, USA; hakob.kocharyan@moffitt.org (H.K.); mustafa.al-roubaie@moffitt.org (M.A.-R.); 4Department of Vascular and Interventional Radiology, David Grant USAF Medical Center, Fairfield, CA 94535, USA; christopher.d.yeisley.mil@health.mil

**Keywords:** adrenal hemorrhage, periduodenal abscess, CT-guided abscess drainage

## Abstract

A spontaneous adrenal hemorrhage is a rare complication occurring in the setting of interventional radiology procedures. Here, we present the case of a 73-year-old male who underwent CT-guided drainage of a periduodenal collection. During the procedure, he developed sudden onset left back pain and hypertension, which revealed a left adrenal gland hemorrhage on CT imaging. Potential mechanisms for this complication include a physiological stress response, procedure-associated hypercoagulability, and direct trauma to the adrenal gland. Moreover, an adrenal hemorrhage should be monitored closely as it can lead to severe clinical consequences requiring treatment with IV analgesics and antihypertensives.

A spontaneous adrenal hemorrhage is a rare but life-threatening complication that, if left untreated, can lead to states of adrenal excess or adrenal insufficiency [1]. There are a variety of etiologies for an adrenal hemorrhage, including stress, infection, trauma, neoplasia, bleeding disorders, and complications of anticoagulation [2]. Although the diagnosis of an adrenal hemorrhage can be challenging due to a non-specific clinical presentation, determining the etiology of this condition is beneficial in guiding the treatment plan. Here, we present the case of a spontaneous unilateral adrenal hemorrhage during CT-guided periduodenal abscess drainage. We also review the current literature on potential mechanisms for this rare complication.

A 73-year-old male with a past medical history of ampullary neoplasm status post endoscopic ultrasound with fine needle aspiration and plastic biliary stent placement presents with right upper quadrant (RUQ) pain, elevated bilirubin, and leukocytosis two days after receiving endoscopic retrograde cholangiopancreatography (ERCP). Computed tomography (CT) of the abdomen revealed moderate free fluid adjacent to the extrahepatic biliary system, pancreatic head, and proximal duodenum (Figure 1). Blood cultures were negative. Repeat ERCP was performed, which showed biliary stent obstruction. The stent was removed and replaced with a 10 mm × 40 mm bare metal stent. Three days after the repeat ERCP, the patient reported worsening abdominal pain and CT showed an increase in RUQ simple fluid. A hepatobiliary iminodiacetic acid (HIDA) scan was performed to rule out biliary leak, which was negative. Due to the patient’s worsening leukocytosis, a repeat CT was performed which showed interval increase, organization, and hyperenhancement of a periduodenal collection. The patient therefore underwent CT-guided drainage of the collection with moderate sedation, which ultimately grew yeast (Figure 2a). 

During the CT-guided drainage, after the needle punctured the collection and a wire was placed, the patient developed sudden onset left back pain and hypertension. CT imaging revealed a spontaneous left adrenal gland hemorrhage (Figure 2b). The drainage procedure was completed by inserting a 10 French drain over the wire and formed within the central aspect of the collection; turbid fluid was aspirated (Figure 3). After the procedure, the patient was observed in the CT suite. Intravenous (IV) analgesia was administered resulting in pain relief, and serial CT imaging with IV contrast showed stability of the left adrenal hemorrhage. Intravenous antihypertensives were administered to manage the acute hypertension. The patient was subsequently hospitalized for pain and hypertension management, secondary to spontaneous adrenal hemorrhage.

Notable etiologies of an adrenal hemorrhage include trauma, anticoagulation-related hemorrhage, acute stress related to major surgeries, pregnancy, labor, and sepsis, among others [3]. While there have not been any documented cases of an adrenal hemorrhage following an interventional radiology procedure, it has been seen in the setting of surgery including colorectal procedures [4], hip arthroplasty [5], liver transplant [6], and nephrectomy, among others [7]. This case presents a novel incident of an adrenal hemorrhage occurring in the setting of an interventional radiology procedure, specifically the drainage of a periduodenal collection. 

The diagnosis of an adrenal hemorrhage can be difficult due to its non-specific clinical presentation. Suspicion of an adrenal hemorrhage should be raised in a patient with hemodynamic instability accompanied by inexplicable abdominal pain [3]. The range of presentations for an adrenal hemorrhage is wide, with a unilateral hemorrhage often presenting as an asymptomatic to bilateral hemorrhage causing primary adrenal insufficiency—a potentially fatal outcome [3]. The diagnosis is typically confirmed incidentally on CT or MRI, which typically demonstrate a hematoma within the adrenal capsule, and a potential rupture into the retroperitoneal space [3]. 

One possible pathogenesis of this patient’s unilateral adrenal hemorrhage is a physiologic stress response during the procedure. Physiological stressors result in a surge of ACTH with the purpose of stimulating cortisol production and release from the adrenal cortex, as well as increasing adrenal arterial blood flow. Simultaneously, in the stress response, catecholamines are released from the adrenal medulla into adrenal venules, causing venoconstriction and reduced venous drainage. These catecholamines may also stimulate platelet aggregation in the adrenal vein, leading to an adrenal vein thrombosis that obstructs and further reduces venous drainage. This simultaneous increase in arterial supply and decrease in venous drainage causes blood to accumulate in the adrenal gland, potentially leading to rupture and hemorrhage [3,8]. The above mechanism draws a connection between the patient experiencing a stress response during the procedure, causing a hormonal surge leading to a unilateral adrenal hemorrhage.

A second proposed mechanism is procedure-associated hypercoagulability. It is well-established that surgery and other invasive procedures can lead to a post-operative hypercoagulable state, potentially leading to thrombosis [9]. Post-operative thrombosis may occur in the adrenal vein, leading to upstream venous and parenchymal congestion of the adrenal gland. Elevated adrenal pressure can lead to rupture and a subsequent hemorrhage, which may have manifested in this patient. 

The patient’s history of ampullary neoplasm may have also been a contributory factor to spontaneous hemorrhage during the procedure. As one of the most prothrombotic malignancies, pancreatic cancer is associated with a high risk of venous thromboembolism (VTE). One retrospective study found a VTE incidence rate of 18% in pancreatic cancer patients, 80% of which occurred in advanced stages of disease [10]. This suggests the importance of close peri-procedural monitoring and is a possible indication for thromboprophylaxis in high-risk individuals such as this patient. 

Direct trauma to the adrenal gland is worth considering as a potential mechanism for an intraoperative adrenal hemorrhage. This is not probable in the case of this patient despite needle access into the periduodenal space, as the adrenal hemorrhage occurred contralaterally to the site of the needle puncture.

The patient’s sudden onset of hypertension suggests that he experienced symptoms of primary hyperadrenalism due to an adrenal hemorrhage. Although a rare complication, there are a variety of treatments available that may be utilized to manage these symptoms. Interventional options include adrenalectomy and super-selective adrenal arterial embolization, which has been found to reduce blood pressure and may potentially decrease bleeding complications following an adrenal hemorrhage, given its embolic effects [11,12]. The mainstay of treatment for an adrenal hemorrhage is medical management, which includes antihypertensives such as mineralocorticoid receptor antagonists to control hypokalemia and hypertension. While symptoms of an adrenal hemorrhage may be equivocal, this endocrine emergency requires timely intervention to avoid morbid outcomes.

Regardless of its pathogenesis, a post-operative adrenal hemorrhage—either unilateral or bilateral—is a potential complication of IR procedures that should be monitored closely. An adrenal hemorrhage can lead to severe clinical consequences, including acute treatment-resistant hypertension and abdominal pain due to massive release of hormones. Treatment with intravenous analgesics and antihypertensives may be required. Chronically, as the adrenal gland’s production of hormones is stunted, the clinical presentation can reflect adrenal insufficiency and multiple organ failure if left untreated [13]. Treatment with long-term steroid supplementation may be required. Many patients may be asymptomatic after an adrenal hemorrhage, especially if unilateral [13]. With such a diversity of clinical manifestations and the potential to lead to severe homeostatic decompensation, it is critical for physicians to recognize and immediately treat a post-operative adrenal hemorrhage when it occurs.

## Figures and Tables

**Figure 1 diagnostics-14-00334-f001:**
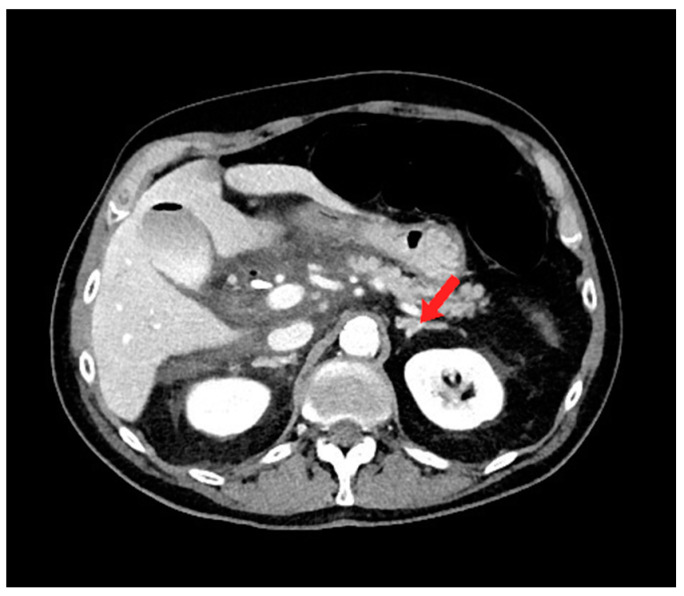
Axial computed tomography (CT) image shows moderate free fluid adjacent to the extrahepatic biliary system, pancreatic head, and proximal duodenum. Note the normal appearance of the left adrenal gland (red arrow).

**Figure 2 diagnostics-14-00334-f002:**
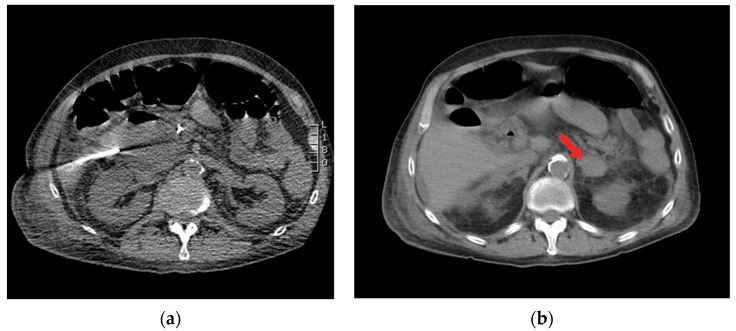
Axial computed tomography (CT) imaging shows (**a**) percutaneous insertion of a 19 Gauge needle for drainage of periduodenal collection, and (**b**) left-sided adrenal hematoma with surrounding stranding of peri-adrenal fat (red arrow).

**Figure 3 diagnostics-14-00334-f003:**
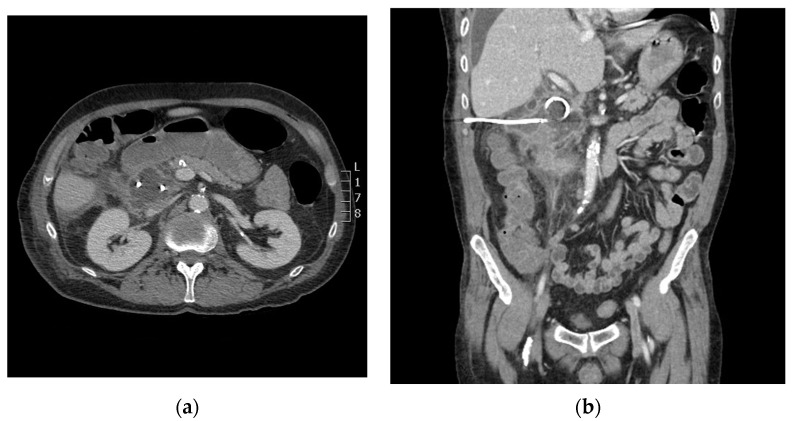
Axial (**a**) and coronal (**b**) computed tomography (CT) imaging shows periduodenal drain placement with a 10 French catheter.

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
