# Peer review of "Spontaneous Contralateral Adrenal Hemorrhage during Periduodenal Abscess Drain Placement"

_diagnostics, 2024, doi:10.3390/diagnostics14030334_

Round 1
Reviewer 1 Report
Comments and Suggestions for Authors
Not very novel but well presented. Not very interesting images but worth to be reported. Better to also mention about the timing of spontaneous hemmorhage with different mechanism. This will help others to understand better the different mechanism and the timing of better supervision.
Author Response
Thank you for your suggestions. We included points about the mechanism of adrenal hemorrhage and the importance of both periprocedural supervision and timely intervention to avoid fatal outcomes.
Reviewer 2 Report
Comments and Suggestions for Authors
The article, titled "Spontaneous Unilateral Adrenal Hemorrhage During Periduodenal Abscess Drain Placement," discusses a rare case of adrenal hemorrhage that occurred spontaneously in a 73-year-old male patient during a CT-guided drainage procedure for a periduodenal collection. The article explores potential mechanisms for this complication, including physiological stress response, procedure-associated hypercoagulability, and direct trauma to the adrenal gland. However, the latter was deemed improbable in this case due to the contralateral occurrence of the hemorrhage relative to the needle puncture site. The article provides a detailed case report, including the patient's medical history, the progression of his condition, the interventional procedure, and the subsequent management of the adrenal hemorrhage. It also reviews the literature on adrenal hemorrhage, its etiologies, clinical presentations, and the importance of promptly recognizing and treating this condition to prevent severe clinical consequences.
Potential limitations include:
1. The article is a single case report, which inherently limits the generalizability of its findings. Case reports typically focus on rare or unusual cases, and their findings may not apply to the broader patient population.
2. The article must comprehensively review the pathogenesis, diagnosis, and management of adrenal hemorrhage. While it does discuss potential mechanisms for the observed complication, it does not delve into the broader context of adrenal hemorrhage as a clinical condition.
3. The article does not discuss the patient's follow-up or long-term outcomes. This information could provide valuable insights into similar cases' prognosis and long-term management.
4. The article does not discuss alternative management strategies or compare the chosen management approach with other potential approaches. This could limit its usefulness for clinicians seeking guidance on managing similar cases.
5. The article must provide a detailed discussion of the literature on adrenal hemorrhage. While it does mention some relevant studies, a more comprehensive literature review could provide a more robust context for the reported case.
6. The article does not discuss the potential role of the patient's underlying medical conditions (e.g., ampullary neoplasm) in the development of the adrenal hemorrhage. This could be a significant omission, as underlying conditions can often influence the course and management of acute medical complications.
Comments on the Quality of English Languageminor
Author Response
Thank you for your suggestions. While we agree that as a single case report this reduces its generalizability, we believe that the patient’s presentation is unique enough to warrant the sharing of our findings. Per your suggestions, we have included a more comprehensive review of the pathogenesis, diagnosis, and management of adrenal hemorrhage. Unfortunately, information about the patient’s follow-up is limited which impacts the ability to discuss long-term management. We included further discussion of alternative management strategies based on the patient’s presenting symptoms. We also added more information about the potential role of ampullary neoplasm in the development of adrenal hemorrhage.
Reviewer 3 Report
Comments and Suggestions for Authors
This is a report of a rare condition of spontaneous hemorrhage of the adrenal gland.
I think a better title would be "contralateral adrenal Hemorrhage during periduodenal abscess drain placement." Contralateral would be better, not Unilateral.
It may be better to arrange the images so that they can be viewed at the same time, as it is difficult to think when the images change pages.
Author Response
Thank you for your suggestions. We have adjusted the title to say “contralateral” rather than “unilateral”. We also arranged the images so that they can be viewed at the same time.
Round 2
Reviewer 2 Report
Comments and Suggestions for Authors
i am satisfied with revision
Comments on the Quality of English Languageminor